# New Explicit Propagating Solitary Waves Formation and Sensitive Visualization of the Dynamical System

Rana Muhammad Zulqarnain [1], Wen-Xiu Ma [1,2,3,4,*], Sayed M. Eldin [5], Khush Bukht Mehdi [6] and Waqas Ali Faridi [6]

1. College of Mathematics and Computer Science, Zhejiang Normal University, Jinhua 321004, China
2. Department of Mathematics, King Abdulaziz University, Jeddah 21589, Saudi Arabia
3. Department of Mathematics and Statistics, University of South Florida, Tampa, FL 33620-5700, USA
4. School of Mathematical and Statistical Sciences, North-West University, Mafikeng Campus, Private Bag X2046, Mmabatho 2735, South Africa
5. Center of Research, Faculty of Engineering, Future University in Egypt, New Cairo 11835, Egypt
6. Department of Mathematics, University of Management and Technology, Lahore 54770, Pakistan
* Correspondence: wma3@usf.edu

**Abstract:** This work discusses the soliton solutions for the fractional complex Ginzburg–Landau equation in Kerr law media. It is a particularly fascinating model in this context as it is a dissipative variant of the Hamiltonian nonlinear Schrödinger equation with solutions that create localized singularities in finite time. The $\phi^6$-model technique is one of the generalized methodologies exerted on the fractional complex Ginzburg–Landau equation to find the new solitary wave profiles. As a result, solitonic wave patterns develop, including Jacobi elliptic function, periodic, dark, bright, single, dark-bright, exponential, trigonometric, and rational solitonic structures, among others. The assurance of the practicality of the solitary wave results is provided by the constraint condition corresponding to each achieved solution. The graphical 3D and contour depiction of the attained outcomes is shown to define the pulse propagation behaviors while imagining the pertinent data for the involved parameters. The sensitive analysis predicts the dependence of the considered model on initial conditions. It is a reliable and efficient technique used to generate generalized solitonic wave profiles with diverse soliton families. Furthermore, we ensure that all results are innovative and mark remarkable impacts on the prevailing solitary wave theory literature.

**Keywords:** exact solitary wave structures; Jacobi elliptic functions; fractional complex Ginzburg–Landau equation; $\phi^6$-model expansion method; beta derivative; sensitive analysis

## 1. Introduction

Exploring exact and solitary traveling wave solutions for nonlinear partial differential equations plays a vital role in nonlinear physical phenomena. Nonlinear wave phenomena occur in many scientific and engineering fields, such as plasma physics, fluid mechanics, biology, optical fiber, solid-state physics, chemical physics, chemical kinematics, and geochemistry. Nonlinear wave phenomena such as dissipation, dispersion, reaction, diffusion, and convection should be included in the nonlinear wave equation. Over the past few eras, innovative exact solutions may support a determination of novel phenomena [1–4]. Additionally, soliton theory has captivated considerable devotion in experimental exploration by scientific societies and intellectuals, as it is a dynamic sector of investigation in broadcastings, engineering, mathematical physics, and numerous other divisions of nonlinear discipline. Particularly, solitons have been extensively studied in the present era. Solitons are types of solitary waves that propagate waves deprived of being scattered over vast distances, i.e., they retain their figure over long distances. Solitons are the major strategy for a telecommunication society. Because of this feature, they are of phenomenal reputation in nonlinear science. Soliton replicas have many purposes, such as solitary

wave-based communication contacts, fiber amplifiers, optical pulse compressors, etc. Soliton theory has led to research for investigators due to its application in assorted arenas such as broadcasting, enterprise, statistical materials science, mathematical physics, and various parts of nonlinear problems [5–8].

Over the past two centuries, fractional calculus has attracted the attention of various intellectuals. Use them to model numerous nonlinear facets, including biological, fluid, and chemical processes. Fractional order partial differential equations (PDEs) are generalizations of conventional order PDEs. The literature comprises numerous descriptions of fractional derivatives, such as the Hadamard derivative [9], the Weyl derivative [10], the Riesz derivative [11], He's fractional derivative [12], Riemann–Liouville [13,14], Abel–Riemann derivative [15], Caputo [16], Caputo–Fabrizio [17] Atangana–Baleanu derivative in the perspective of Caputo [18], the conformable fractional derivative [19], the innovative truncated M-fractional derivative [20]. Atangana et al. [21] recently developed the beta derivative that contains many of the properties considered to be the confines of fractional derivatives. This derivative has stimulating effects in various fields, such as optical physics, circuit analysis, chaos theory, fluid mechanics, disease analysis, biological modeling, etc.

Several models are currently being considered for soliton solutions [22–29]. One model that has been under contemplation for several years is the complex Ginzburg–Landau (CGL). The complex Ginzburg–Landau equation (CGLE) is one of the best models to define optical phenomena [30–32]. To better study complex optical phenomena and their nature, the finest approaches are to bargain exact traveling solutions to CGLE that designate nonlinear optical phenomena. Numerous potent mathematical methods have recently been used to obtain exact soliton solutions to CGLE. Liu et al. [33] obtained the kink and periodic wave solutions by using the Hirota bilinear method. Inc et al. [34] obtained the bright and singular soliton solutions for the non-linearity term of the CGL model by utilizing the Sine-Gordon method. Arnous et al. [35] observed the optical soliton solution by utilizing the improved simple equation technique. The quadratic and multiple solitons of n-dimension CGLE were extended by Khater et al. [36] using the Sine-Gordon expansion method. Das et al. [37] used the *F*-expansion method to obtain bright and dark solitons of CGLE.

The current research sheds light on the space-time fractional CGLE [38,39]. The space-time fractional CGL model associated here is deliberated by

$$
\begin{aligned}
& i {}_0^A D_t^\alpha u + a {}_0^A D_x^{2\alpha} u + cH(|u|^2)u \\
& = \frac{1}{|u|^2 u*}\left\{\delta {}_0^A D_x^{2\alpha}(|u|^2)|u|^2 - N\left({}_0^A D_x^\alpha(|u|^2)\right)^2\right\} + Pu,
\end{aligned}
\tag{1}
$$

where $\alpha$ and $\beta$ are the fractional parameters, $x$ signifies distance through the fiber, $t$ represents time in dimensionless form $a$, $c$, and $P$ are valued constants. The sign $*$ shows the complex conjugate of the function $u(x, t)$ and $H$ is a real-valued algebraic function, and its consistency is organized by a complex function $H(|u|^2)u : C \to C$. Now, $C$ a two-dimensional linear space $R^2$, $H(|u|^2)u$ is $k$ times continuously differentiable real-valued function [40]:

$$
H\left(|u|^2\right)u \in \overset{\infty}{\underset{p,q=1}{\tilde{U}}} C^k\left((-q, q) \times (-p, p); R^2\right),
\tag{2}
$$

where $\delta = 2N$, then Equation (1) reduces to

$$
\begin{aligned}
& i {}_0^A D_t^\alpha u + a {}_0^A D_x^{2\alpha} u + cH(|u|^2)u \\
& = \frac{N}{|u|^2 u*}\left\{2 {}_0^A D_x^{2\alpha}(|u|^2)|u|^2 - \left({}_0^A D_x^\alpha(|u|^2)\right)^2\right\} + Pu.
\end{aligned}
\tag{3}
$$

Equation (2) is one of many models that control the dynamics of fiber optic pulse diffusion at transcontinental and transoceanic distances. Sulaiman et al. [41] extended the Sine-Gordon expansion method and intended the conformable time-space fractional CGLE. Abdo et al. [42] deliberated the fractional CGLE using the extended Jacobi elliptic function

expansion system. Arshed [43] utilized the $\exp(-\phi(\xi))-$ expansion method and built the soliton solutions to fractional CGLE. Ghanbari and Gomez-Aguilar [44] use exponential rational functional methods to explore periodic and hyperbolic soliton solutions CGLE. Lu et al. [45] considered the (2+1) dimensional fractional CGLE by fractional Riccati and bifurcation methods. Hussain and Jhangeer [46] obtained the optical solitons of fractional CGLE with conformable, beta, and M-truncated derivatives. Akram et al. [47] studied the optical solitons for the fractional CGLE with Kerr law non-linearity engaging diverse fractional differential operators. Sadaf et al. [48] obtained the dark, bright, complexion, singular and periodic optical solitons of fractional CGLE with Kerr law non-linearity implementing conformable, beta, and M-truncated derivatives. Zafar et al. [49] used improved exp-function and the Kudryshov method to obtain kink, bright, W-shaped bright and dark solitons for fractional CGL models. This model was confirmed with quadratic-cubic laws, Kerr's law, and parabolic laws of nonlinear fibers.

The main focus of this investigation is to use the new implication of fractional-order derivatives, such as beta fractional derivatives [21] for space-time fractional CGLE [38,39], and to determine the novel composite exact traveling wave solutions in terms of light, dark, singular soliton, and periodic solitary wave solutions with Kerr's law using the $\phi^6$-model expansion method [50–54]. To the best of our familiarity, the solutions attained are broader and in diverse arrangements, which have not been stated in earlier available studies [38,42–49]. Moreover, we attain the dynamic behavior of a solitary wave solution (SWS) involving a class of Jacobi elliptic functions under the constraints. Due to its imperative solicitation in nonlinear optics, this solution is significant for advanced studies of this model.

The rest of the paper is organized as follows. In Section 2, the beta derivative and its properties are deliberated, and the techniques of $\phi^6$-model expansion scheme is discussed in Section 3. In Section 4, $\phi^6$-model expansion model is utilized for the space-time fractional CGLE. The graphical assessments of our attained solutions are signified in 3D and contour plots for multiple values of parameters in Section 5. Section 6 includes the study of the sensitivity analysis. Finally, conclusions are publicized in Section 7.

## 2. Beta-Derivative and Its Properties

**Definition:** *Suppose a function $h(x)$ that is defined $\forall$ as non-negative $x$. Therefore, the beta derivative of the function $h(x)$ is given as [21]:*

$$_0^A D_x^\beta (h(x)) = \lim_{\varepsilon \to 0} \frac{h\left(x + \varepsilon\left(x + \frac{1}{\Gamma(\beta)}\right)^{1-\beta}\right) - h(x)}{\varepsilon}, \quad 0 < \beta \le 1.$$

*Properties: Assuming that $a$ and $b$ are real numbers, $g(x)$ and $h(x)$ are two functions $\beta-$ differentiable and $\beta \in (0,1]$ then.*

$$
\begin{aligned}
&i. \ _0^A D_x^\beta (ag(x) + bh(x)) = a_0^A D_x^\beta (g(x)) + b_0^A D_x^\beta (h(x)), \ \forall a, b \ \in \ R. \\
&ii. \ _0^A D_x^\beta (c) = 0, \ \text{for any constant } c. \\
&iii. \ _0^A D_x^\beta (g(x)h(x)) = h(x)_0^A D_x^\beta (g(x)) + g(x)_0^A D_x^\beta (h(x)). \\
&iv. \ _0^A D_x^\beta \left(\frac{g(x)}{h(x)}\right) = \frac{h(x)_0^A D_x^\beta (g(x)) + g(x)_0^A D_x^\beta (h(x))}{(h(x))^2}. \\
&v. \ _0^A D_x^\beta (g(x)) = \left(x + \frac{1}{\Gamma(\beta)}\right)^{1-\beta} \frac{dg(x)}{dx}.
\end{aligned}
\tag{4}
$$

## 3. Representation of the $\phi^6$-Model Expansion Method

Suppose that the nonlinear (PDE) is defined as:

$$F(u,\ u_x,\ u_t,\ u_{xx},\ u_{tt}, \ldots\ldots) = 0, \tag{5}$$

Here, $u(x,t)$, partial derivatives $F$ as a polynomial.
The core phases of this scheme are:
Step 1: By the subsequent transformation

$$u(x,t) = U(\eta), \quad \eta = x - vt. \tag{6}$$

where $v$ is the wave speed, and the PDE is converted into the following ODE.

$$G(U,\ U\prime,\ U'',\ U''', \ldots) = 0, \tag{7}$$

At this stage, $G$ is a polynomial and

$$U = U(\eta), \quad U\prime = \frac{dU}{d\eta}, \quad U'' = \frac{d^2U}{d\eta^2}, \quad U''' = \frac{d^3U}{d\eta^3}, \quad \cdots.$$

Step 2: Suppose that Equation (7) has the formal solution:

$$U(\eta) = \sum_{i=0}^{2M} a_i \phi^i(\eta), \tag{8}$$

where $a_i(i = 0,\ 1,\ 2,\ \cdots,\ 2M)$ are constants to be resolved later, while $\phi(\eta)$ satisfies the well-known auxiliary nonlinear ODE.

$$\begin{aligned}\phi\prime^2(\eta) &= h_0 + h_2\phi^2(\eta) + h_4\phi^4(\eta) + h_6\phi^6(\eta),\\ \phi''(\eta) &= h_2\phi(\eta) + 2h_4\phi^3(\eta) + 3h_6\phi^5(\eta),\end{aligned} \tag{9}$$

where $h_i(i = 0,\ 2,\ 4,\ 6)$ are real constants.
Step 3: We govern the positive integer $N$ in Equation (8) by balancing the highest-order derivative with the highest nonlinear terms in Equation (7).
Step 4: It is well known [52–54] that Equation (9) has the solution

$$\phi(\eta) = \frac{\Omega(\eta)}{\sqrt{f\Omega^2(\eta) + g}}, \tag{10}$$

where $(f\Omega^2(\eta) + g) > 0$ and $\Omega(\eta)$ is the solution of the Jacobian elliptic equation.

$$\Omega\prime^2 = l_0 + l_2\Omega^2(\eta) + l_4\Omega^4(\eta), \tag{11}$$

where $l_j(j = 0,2,4)$ are constants to be determined later, while $f$ and $g$ are given by

$$f = \frac{h_4(l_2 - h_2)}{(l_2 - h_2)^2 + 3l_0l_4 - 2l_2(l_2 - h_2)}, \tag{12}$$

$$g = \frac{3l_0h_4}{(l_2 - h_2)^2 + 3l_0l_4 - 2l_2(l_2 - h_2)}, \tag{13}$$

Under the constraints condition

$$h_4{}^2(l_2 - h_2)(9l_0l_4 - (l_2 - h_2)(2l_2 + h_2)) + 3h_6\left(3l_0l_4 - \left(l_2{}^2 - h_2{}^2\right)\right)^2 = 0. \tag{14}$$

Step 5: Equation (11) has the Jacobi elliptic solution defined in table as

| No. | $l_0$ | $l_2$ | $l_4$ | $U(\eta)$ |
|---|---|---|---|---|
| 1 | $1$ | $-(1+m^2)$ | $m^2$ | $sn(\eta)$ or $cd(\eta)$ |
| 2 | $1-m^2$ | $2m^2-1$ | $-m^2$ | $cn(\eta)$ |
| 3 | $m^2-1$ | $2-m^2$ | $-1$ | $dn(\eta)$ |
| 4 | $m^2$ | $-(1+m^2)$ | $1$ | $ns(\eta)$ or $dc(\eta)$ |
| 5 | $-m^2$ | $2m^2-1$ | $1-m^2$ | $nc(\eta)$ |
| 6 | $-1$ | $2-m^2$ | $-(1-m^2)$ | $nd(\eta)$ |
| 7 | $1$ | $2-m^2$ | $1-m^2$ | $sc(\eta)$ |
| 8 | $1$ | $2m^2-1$ | $-m^2(1-m^2)$ | $sd(\eta)$ |
| 9 | $1-m^2$ | $2-m^2$ | $1$ | $cs(\eta)$ |
| 10 | $-m^2(1-m^2)$ | $2m^2-1$ | $1$ | $ds(\eta)$ |
| 11 | $\frac{1-m^2}{4}$ | $\frac{1+m^2}{2}$ | $\frac{1-m^2}{4}$ | $nc(\eta)\pm sc(\eta)$ or $\frac{cn(\eta)}{1\pm sn(\eta)}$ |
| 12 | $\frac{-(1-m^2)^2}{4}$ | $\frac{1+m^2}{2}$ | $\frac{-1}{4}$ | $mcn(\eta)\pm dn(\eta)$ |
| 13 | $\frac{1}{4}$ | $\frac{1-2m^2}{2}$ | $\frac{1}{4}$ | $\frac{sn(\eta)}{1\pm cn(\eta)}$ |
| 14 | $\frac{1}{4}$ | $\frac{1+m^2}{2}$ | $\frac{(1-m^2)^2}{4}$ | $\frac{sn(\eta)}{cn(\eta)\pm dn(\eta)}$ |

Now, we define Jacobian elliptic functions with their limitations to derive the exact solutions of this method which are given in the following table.

| Function | $m\to 1$ | $m\to 0$ | Function | $m\to 1$ | $m\to 0$ |
|---|---|---|---|---|---|
| $sn(\eta,m)$ | $\tanh(\eta)$ | $\sin(\eta)$ | $ns(\eta,m)$ | $\coth(\eta)$ | $\csc(\eta)$ |
| $cd(\eta)$ | $1$ | $\cos(\eta)$ | $dc(\eta)$ | $1$ | $\sec(\eta)$ |
| $cn(\eta)$ | $\text{sech}(\eta)$ | $\cos(\eta)$ | $nc(\eta)$ | $\cosh(\eta)$ | $\sec(\eta)$ |
| $dn(\eta)$ | $\text{sech}(\eta)$ | $1$ | $nd(\eta)$ | $\cosh(\eta)$ | $1$ |
| $sc(\eta)$ | $\sinh(\eta)$ | $\tan(\eta)$ | $cs(\eta)$ | $\text{csch}(\eta)$ | $\cot(\eta)$ |
| $sd(\eta)$ | $\sinh(\eta)$ | $\sin(\eta)$ | $ds(\eta)$ | $\text{csch}(\eta)$ | $\csc(\eta)$ |

## 4. Appliance of $\phi^6$-Model Expansion Method

By using the traveling wave transformation

$$u(x,t)=U(\eta)e^{i\theta(x,t)},\ \eta=\tfrac{1}{\alpha}\left(x+\tfrac{1}{\Gamma(\alpha)}\right)^\alpha-\tfrac{v}{\alpha}\left(t+\tfrac{1}{\Gamma(\alpha)}\right)^\alpha.$$
$$\Theta(x,t)=-\tfrac{k}{\alpha}\left(x+\tfrac{1}{\Gamma(\alpha)}\right)^\alpha+\tfrac{w}{\alpha}\left(t+\tfrac{1}{\Gamma(\alpha)}\right)^\alpha+\theta_0(\varepsilon),$$
$$\tag{15}$$

where $u(x,t)$, $w$, $k$, $v$, $\Theta(x,t)$ and $\theta_0(\varepsilon)$ represents the pulse shape, wave number, frequency, speed, phase component, and phase function of soliton, respectively.

Substituting Equation (15) into Equation (3), an ODE is attained, whose real and imaginary parts, respectively, are:

$$(a-4N)U''-\left(w+ak^2+P\right)U+cH\left(U^2\right)U=0,\tag{16}$$

And

$$v=-2ak.\tag{17}$$

Now, the research concentration is to contemplate Equation (16) with the shape of nonlinear fibers, i.e., Kerr law.

*Kerr Law*

In this case, when we take

$$H(U) = U. \tag{18}$$

This appears in water waves and in nonlinear fiber optics Biswas et al. [55]. Then, Equation (3) becomes:

$$
\begin{aligned}
&i\,{}_0^A D_t^\alpha u + a\,{}_0^A D_x^{2\alpha} u + c\left(|u|^2\right)u \\
&= \frac{N}{|u|^2 u *}\left\{2\,{}_0^A D_x^{2\alpha}\left(|u|^2\right)|u|^2 - \left({}_0^A D_x^\alpha\left(|u|^2\right)\right)^2\right\} + Pu.
\end{aligned}
\tag{19}
$$

Thus, Equation (19) changes to

$$(a - 4N)U'' - \left(w + ak^2 + P\right)U + cU^3 = 0. \tag{20}$$

According to the balance principle, we obtain $m = 1$. Putting $m = 1$ into Equation (8), we then get

$$U(\eta) = a_0 + a_1\phi(\eta) + a_2\phi^2(\eta). \tag{21}$$

Here, $a_0$, $a_1$ and $a_1$ are unknown parameters. Now, substitute Equation (9) along with Equation (13) into Equation (20) and compare the polynomial coefficients equal to zero.

We acquired:

$$
\begin{aligned}
\phi^0(\eta) \quad &: a_0{}^3 c + 2aa_2 h_0 - aa_0 k^2 - 8a_2 h_0 N - a_0 P - a_0 w = 0, \\
\phi^1(\eta) \quad &: 3a_0{}^2 a_1 c + aa_1 h_2 - aa_1 k^2 - 4a_1 h_2 N - a_1 P - a_1 w = 0, \\
\phi^2(\eta) \quad &: 3a_0 a_1{}^2 c + 3a_0{}^2 a_2 c + 2aa_2 h_1 + 2aa_2 h_2 - aa_2 k^2 \\
&\quad -8a_2 h_1 N - 8a_2 h_2 N - a_2 P - a_2 w = 0, \\
\phi^3(\eta) \quad &: a_1{}^3 c + 6a_0 a_1 a_2 c + 2aa_1 h_4 - 8a_1 h_4 N = 0, \\
\phi^4(\eta) \quad &: 3a_1{}^2 a_2 c + 3a_0 a_2{}^2 c + 6aa_2 h_4 - 24a_2 h_4 N = 0, \\
\phi^5(\eta) \quad &: 3a_1 a_2{}^2 c + 3aa_1 h_6 - 12a_1 h_6 N = 0, \\
\phi^6(\eta) \quad &: a_2{}^3 c + 8aa_2 h_6 - 32a_2 h_6 N = 0.
\end{aligned}
\tag{22}
$$

Mathematica software is used to resolve the system (22) and obtain a set of solutions,

$$
\begin{aligned}
&a_0 = a_0, \ a_1 = 0, \ a_2 = a_2, \\
&h_0 = \frac{a_0 \Delta}{2a_2 K}, \ h_2 = \frac{\Delta_1}{4K}, \ h_4 = -\frac{a_0 a_2 c}{2K}, \ h_6 = \frac{-a_2{}^2 c}{8K},
\end{aligned}
\tag{23}
$$

here $K = a - 4N$, $\Delta = -a_0{}^2 c + ak^2 + P + w$ and $\Delta_1 = -3a_0{}^2 c + ak^2 + P + w$.

The exact solutions of Equation (3) are:

**Result 1**

If $l_0 = 1$, $l_2 = -\left(1 + m^2\right)$, $l_4 = m^2, 0 < m < 1$, then $\Omega(\eta) = sn(\eta)$ thus, we have

$$U_1 = \left(a_0 + a_2\left(\frac{sn^2(\eta)}{f sn^2(\eta) + g}\right)\right)e^{i\theta(x,t)}, \tag{24}$$

where $f$ and $g$ are given as

$$
\begin{aligned}
f &= \frac{h_4\left(-m^2 - h_2 - 1\right)}{\left(-m^2 - h_2 - 1\right)^2 - 2\left(-m^2 - 1\right)\left(-m^2 - h_2 - 1\right) + 3m^2}, \\
g &= \frac{3h_4}{\left(-m^2 - h_2 - 1\right)^2 - 2\left(-m^2 - 1\right)\left(-m^2 - h_2 - 1\right) + 3m^2},
\end{aligned}
\tag{25}
$$

When $m \to 1$, $\Omega(\eta) = sn(\eta) = \tanh(\eta)$, we can acquire an SWS.

$$U_{1,1} = \left( a_0 + a_2 \left( \frac{\tanh^2(\eta)}{f \tanh^2(\eta) + g} \right) \right) e^{i\theta(x,t)}, \tag{26}$$

or $\Omega(\eta) = cd(\eta) = 1$, we can acquire an SWS.

$$U_{1,2} = \left( a_0 + a_2 \left( \frac{\Omega(\eta)^2}{f \Omega(\eta)^2 + g} \right) \right) e^{i\theta(x,t)}, \tag{27}$$

When $m \to 0$, $\Omega(\eta) = sn(\eta) = \sin(\eta)$, we can acquire an SWS.

$$U_{1,3} = \left( a_0 + a_2 \left( \frac{\sin^2(\eta)}{f \sin^2(\eta) + g} \right) \right) e^{i\theta(x,t)}, \tag{28}$$

or $\Omega(\eta) = sn(\eta) = \cos(\eta)$, we can acquire an SWS.

$$U_{1,4} = \left( a_0 + a_2 \left( \frac{\cos^2(\eta)}{f \cos^2(\eta) + g} \right) \right) e^{i\theta(x,t)}, \tag{29}$$

under constraints defined as:

$$
\begin{aligned}
& \left( \frac{a_0 a_2 c}{2(a-4N)} \right)^2 \left( -m^2 - \frac{P - a_0^2 c + ak^2 + w}{4(a-4N)} - 1 \right) \\
& \times \left( 9m^2 - \left( -m^2 - \frac{P - a_0^2 c + ak^2 + w}{4(a-4N)} - 1 \right) \left( 2(-m^2 - 1) + \frac{P - a_0^2 c + ak^2 + w}{4(a-4N)} \right) \right) \\
& - \frac{3a_2^2 c}{8(a-4N)} \left( 3m^2 - (-m^2 - 1)^2 + (\frac{P - a_0^2 c + ak^2 + w}{4(a-4N)})^2 \right)^2 = 0.
\end{aligned}
$$

**Result 2**

If $l_0 = 1 - m^2$, $l_2 = 2m^2 - 1$, $l_4 = -m^2, 0 < m < 1$, then $\Omega(\eta) = cn(\eta)$ thus, we have

$$U_2 = \left( a_0 + a_2 \left( \frac{cn^2(\eta)}{f cn^2(\eta) + g} \right) \right) e^{i\theta(x,t)}, \tag{30}$$

where $f$ and $g$ are given as

$$
\begin{aligned}
f &= \frac{h_4 \left( 2m^2 - h_2 - 1 \right)}{(2m^2 - h_2 - 1)^2 - 2(2m^2 - 1)(2m^2 - h_2 - 1) - 3(1 - m^2)m^2}, \\
g &= \frac{3(1 - m^2) h_4}{(2m^2 - h_2 - 1)^2 - 2(2m^2 - 1)(2m^2 - h_2 - 1) - 3(1 - m^2)m^2},
\end{aligned} \tag{31}
$$

When $m \to 1$, $\Omega(\eta) = cn(\eta) = \mathrm{sech}(\eta)$, we can acquire an SWS.

$$U_{2,1} = \left( a_0 + a_2 \left( \frac{\mathrm{sech}^2(\eta)}{f \mathrm{sech}^2(\eta) + g} \right) \right) e^{i\theta(x,t)}, \tag{32}$$

When $m \to 0$, $\Omega(\eta) = sn(\eta) = \cos(\eta)$, we can acquire an SWS.

$$U_{2,2} = \left( a_0 + a_2 \left( \frac{\cos^2(\eta)}{f \cos^2(\eta) + g} \right) \right) e^{i\theta(x,t)}, \tag{33}$$

under constraints defined as

$$\left(\frac{a_0 a_2 c}{2(a-4N)}\right)^2 \left(2m^2 - \frac{P-a_0^2 c + ak^2 + w}{4(a-4N)} - 1\right)$$

$$\times \left(\begin{array}{c} -\left(2m^2 - \frac{P-a_0^2 c + ak^2 + w}{4(a-4N)} - 1\right) \\ \left(2(2m^2 - 1) + \frac{P-a_0^2 c + ak^2 + w}{4(a-4N)}\right) - 9(1-m^2)m^2 \end{array}\right)$$

$$-\frac{3a_2^2 c}{8(a-4N)}\left(-3(1-m^2)m^2 - (2m^2 - 1)^2 + \left(\frac{P-a_0^2 c + ak^2 + w}{4(a-4N)}\right)^2\right)^2 = 0.$$

**Result 3**

If $l_0 = m^2 - 1$, $l_2 = 2 - m^2$, $l_4 = -1$, $0 < m < 1$, then $\Omega(\eta) = dn(\eta)$ thus, we have

$$U_3 = \left(a_0 + a_2 \left(\frac{dn^2(\eta)}{f dn^2(\eta) + g}\right)\right) e^{i\theta(x,t)}, \tag{34}$$

where $f$ and $g$ are given as

$$
\begin{aligned}
f &= \frac{h_4\left(-m^2 - h_2 + 2\right)}{(-m^2 - h_2 + 2)^2 - 2(2 - m^2)(-m^2 - h_2 + 2) - 3(m^2 - 1)}, \\
g &= \frac{3(m^2 - 1)h_4}{(-m^2 - h_2 + 2)^2 - 2(2 - m^2)(-m^2 - h_2 + 2) - 3(m^2 - 1)},
\end{aligned} \tag{35}
$$

When $m \to 1$, $\Omega(\eta) = dn(\eta) = \operatorname{sech}(\eta)$, we can acquire an SWS.

$$U_{3,1} = \left(a_0 + a_2 \left(\frac{\operatorname{sech}^2(\eta)}{f\operatorname{sech}^2(\eta) + g}\right)\right) e^{i\theta(x,t)}, \tag{36}$$

when $m \to 0$, $\Omega(\eta) = dn(\eta) = 1$, we can acquire an SWS.

$$U_{3,2} = \left(a_0 + a_2 \left(\frac{\Omega(\eta)^2}{f\Omega(\eta)^2 + g}\right)\right) e^{i\theta(x,t)}, \tag{37}$$

under constraints defined as

$$\left(\frac{a_0 a_2 c}{2(a-4N)}\right)^2 \left(-m^2 - \frac{P-a_0^2 c + ak^2 + w}{4(a-4N)} + 2\right)$$

$$\times \left(\begin{array}{c} -\left(-m^2 - \frac{P-a_0^2 c + ak^2 + w}{4(a-4N)} + 2\right) \\ \left(2(2-m^2) + \frac{P-a_0^2 c + ak^2 + w}{4(a-4N)}\right) - 9(m^2 - 1) \end{array}\right)$$

$$-\frac{3a_2^2 c}{8(a-4N)}\left(-(2-m^2)^2 - 3(m^2 - 1) + (\frac{P-a_0^2 c + ak^2 + w}{4(a-4N)})^2\right)^2 = 0.$$

**Result 4**

If $l_0 = m^2$, $l_2 = -(m^2 + 1)$, $l_4 = 1$, $0 < m < 1$, then $\Omega(\eta) = ns(\eta)$ or $dc(\eta)$ thus, we have

$$U_4 = \left(a_0 + a_2 \left(\frac{\Omega(\eta)^2}{f\Omega(\eta)^2 + g}\right)\right) e^{i\theta(x,t)}, \tag{38}$$

where $f$ and $g$ are given as

$$
\begin{aligned}
f &= \frac{h_4\left(-m^2 - h_2 - 1\right)}{(-m^2 - h_2 - 1)^2 - 2(-m^2 - 1)(-m^2 - h_2 - 1) + 3m^2}, \\
g &= \frac{3m^2 h_4}{(-m^2 - h_2 - 1)^2 - 2(-m^2 - 1)(-m^2 - h_2 - 1) + 3m^2},
\end{aligned} \tag{39}
$$

when $m \to 1$, $\Omega(\eta) = ns(\eta) = \coth(\eta)$, we can acquire an SWS.

$$U_{4,1} = \left( a_0 + a_2 \left( \frac{\coth^2(\eta)}{f\coth^2(\eta) + g} \right) \right) e^{i\theta(x,t)}, \tag{40}$$

or $\Omega(\eta) = dc(\eta) = 1$, we obtain

$$U_{4,2} = \left( a_0 + a_2 \left( \frac{\Omega(\eta)^2}{f\Omega(\eta)^2 + g} \right) \right) e^{i\theta(x,t)}, \tag{41}$$

when $m \to 0$, $\Omega(\eta) = ns(\eta) = \csc(\eta)$, we can acquire an SWS.

$$U_{4,3} = \left( a_0 + a_2 \left( \frac{\csc^2(\eta)}{f\csc^2(\eta) + g} \right) \right) e^{i\theta(x,t)}, \tag{42}$$

or $\Omega(\eta) = dc(\eta) = \sec(\eta)$, we obtain

$$U_{4,4} = \left( a_0 + a_2 \left( \frac{\sec^2(\eta)}{f\sec^2(\eta) + g} \right) \right) e^{i\theta(x,t)}, \tag{43}$$

under constraint defined as

$$\left( \frac{a_0 a_2 c}{2(a-4N)} \right)^2 \left( -m^2 - \frac{P - a_0^2 c + ak^2 + w}{4(a-4N)} - 1 \right)$$
$$\times \left( 9m^2 - \left( -m^2 - \frac{P - a_0^2 c + ak^2 + w}{4(a-4N)} - 1 \right) \left( 2(-m^2 - 1) + \frac{P - a_0^2 c + ak^2 + w}{4(a-4N)} \right) \right)$$
$$- \frac{3a_2^2 c}{8(a-4N)} \left( 3m^2 - (m^2 - 1)^2 + (\frac{P - a_0^2 c + ak^2 + w}{4(a-4N)})^2 \right)^2 = 0.$$

**Result 5**

If $l_0 = -m^2$, $l_2 = 2m^2 - 1$, $l_4 = 1 - n^2$, $0 < m < 1$, then $\Omega(\eta) = nc(\eta)$, thus, we have

$$U_5 = \left( a_0 + a_2 \left( \frac{nc^2(\eta)}{fnc^2(\eta) + g} \right) \right) e^{i\theta(x,t)}, \tag{44}$$

where $f$ and $g$ are given as

$$\begin{aligned} f &= \frac{h_4(2m^2 - h_2 - 1)}{(2m^2 - h_2 - 1)^2 - 2(2m^2 - 1)(2m^2 - h_2 - 1) - 3(1 - m^2)m^2}, \\ g &= -\frac{3m^2 h_4}{(2m^2 - h_2 - 1)^2 - 2(2m^2 - 1)(2m^2 - h_2 - 1) - 3(1 - m^2)m^2}, \end{aligned} \tag{45}$$

when $m \to 1$, $\Omega(\eta) = nc(\eta) = \cosh(\eta)$, we can acquire an SWS.

$$U_{5,1} = \left( a_0 + a_2 \left( \frac{\cosh^2(\eta)}{f\cosh^2(\eta) + g} \right) \right) e^{i\theta(x,t)}, \tag{46}$$

When $m \to 0$, $\Omega(\eta) = nc(\eta) = \sec(\eta)$, we can acquire an SWS.

$$U_{5,2} = \left( a_0 + a_2 \left( \frac{\sec^2(\eta)}{f\sec^2(\eta) + g} \right) \right) e^{i\theta(x,t)}, \tag{47}$$

under constraint defined as

$$\left(\frac{a_0 a_2 c}{2(a-4N)}\right)^2 \left(2m^2 - \frac{P - a_0{}^2 c + ak^2 + w}{4(a-4N)} - 1\right)$$

$$\times \left(\begin{array}{c} -\left(2m^2 - \frac{P - a_0{}^2 c + ak^2 + w}{4(a-4N)} - 1\right) \\ \left(2(2m^2 - 1) + \frac{P - a_0{}^2 c + ak^2 + w}{4(a-4N)}\right) - 9(1 - m^2)m^2 \end{array}\right)$$

$$-\frac{3a_2{}^2 c}{8(a-4N)}\left(-3(1 - m^2)m^2 - (2m^2 - 1)^2 + \left(\frac{P - a_0{}^2 c + ak^2 + w}{4(a-4N)}\right)^2\right)^2 = 0.$$

where $h_2$, $h_4$ and $h_6$ are given in Equation (23).

**Result 6**

If $l_0 = -1$, $l_2 = 2 - m^2$, $l_4 = -(1 - n^2)$, $0 < m < 1$, then $\Omega(\eta) = nd(\eta)$, thus, we have

$$U_6 = \left(a_0 + a_2\left(\frac{nd^2(\eta)}{fnd^2(\eta) + g}\right)\right)e^{i\theta(x,t)}, \tag{48}$$

where $f$ and $g$ are given as

$$\begin{aligned} f &= \frac{h_4(-m^2 - h_2 + 2)}{(-m^2 - h_2 + 2)^2 - 2(2 - m^2)(-m^2 - h_2 + 2) - 3(m^2 - 1)}, \\ g &= -\frac{3h_4}{(-m^2 - h_2 + 2)^2 - 2(2 - m^2)(-m^2 - h_2 + 2) - 3(m^2 - 1)}, \end{aligned} \tag{49}$$

When $m \to 1$, $\Omega(\eta) = nd(\eta) = \cosh(\eta)$, we can acquire an SWS.

$$U_{6,1} = \left(a_0 + a_2\left(\frac{\cosh^2(\eta)}{f\cosh^2(\eta) + g}\right)\right)e^{i\theta(x,t)}, \tag{50}$$

when $m \to 0$, $\Omega(\eta) = nd(\eta) = 1$, we obtain.

$$U_{6,2} = \left(a_0 + a_2\left(\frac{\Omega(\eta)^2}{f\Omega(\eta)^2 + g}\right)\right)e^{i\theta(x,t)}, \tag{51}$$

under constraint defined as

$$\left(\frac{a_0 a_2 c}{2(a-4N)}\right)^2 \left(-m^2 - \frac{P - a_0{}^2 c + ak^2 + w}{4(a-4N)} + 2\right)$$

$$\times \left(\begin{array}{c} -\left(-m^2 - \frac{P - a_0{}^2 c + ak^2 + w}{4(a-4N)} + 2\right) \\ \left(2(2 - m^2) + \frac{P - a_0{}^2 c + ak^2 + w}{4(a-4N)}\right) - 9(m^2 - 1) \end{array}\right)$$

$$-\frac{3a_2{}^2 c}{8(a-4N)}\left(-(2 - m^2)^2 - 3(m^2 - 1) + \left(\frac{P - a_0{}^2 c + ak^2 + w}{4(a-4N)}\right)^2\right)^2 = 0.$$

where $h_2$, $h_4$ and $h_6$ are given in Equation (23).

**Result 7**

If $l_0 = 1$, $l_2 = 2 - m^2$, $l_4 = (1 - m^2)$, $0 < m < 1$, then $\Omega(\eta) = sc(\eta)$, thus, we have

$$U_7 = \left(a_0 + a_2\left(\frac{sc^2(\eta)}{fsc^2(\eta) + g}\right)\right)e^{i\theta(x,t)}, \tag{52}$$

where $f$ and $g$ are given as

$$\begin{aligned} f &= \frac{h_4(-m^2 - h_2 + 2)}{(-m^2 - h_2 + 2)^2 - 2(2 - m^2)(-m^2 - h_2 + 2) - 3(1 - m^2)}, \\ g &= \frac{3h_4}{(-m^2 - h_2 + 2)^2 - 2(2 - m^2)(-m^2 - h_2 + 2) - 3(1 - m^2)}, \end{aligned} \tag{53}$$

when $m \to 1$, $\Omega(\eta) = sc(\eta) = \sinh(\eta)$, we can acquire an SWS.

$$U_{7,1} = \left( a_0 + a_2 \left( \frac{\sinh^2(\eta)}{f \sinh^2(\eta) + g} \right) \right) e^{i\theta(x,t)}, \tag{54}$$

when $m \to 0$, $\Omega(\eta) = sc(\eta) = \tan(\eta)$, we obtain.

$$U_{7,2} = \left( a_0 + a_2 \left( \frac{\Omega(\eta)^2}{f\Omega(\eta)^2 + g} \right) \right) e^{i\theta(x,t)}, \tag{55}$$

under constraint defined as

$$\left( \frac{a_0 a_2 c}{2(a-4N)} \right)^2 \left( -m^2 - \frac{P - a_0^2 c + ak^2 + w}{4(a-4N)} + 2 \right)$$
$$\times \left( \begin{array}{c} 9(1 - m^2) - \left( -m^2 - \frac{P - a_0^2 c + ak^2 + w}{4(a-4N)} + 2 \right) \\ \left( 2(2 - m^2) + \frac{P - a_0^2 c + ak^2 + w}{4(a-4N)} \right) \end{array} \right)$$
$$- \frac{3 a_2^2 c}{8(a-4N)} \left( -(2 - m^2)^2 - 3(1 - m^2) + (\frac{P - a_0^2 c + ak^2 + w}{4(a-4N)})^2 \right)^2 = 0.$$

where $h_2$, $h_4$ and $h_6$ are given in Equation (23).

**Result 8**

If $l_0 = 1$, $l_2 = 2m^2 - 1$, $l_4 = -m^2(1 - m^2)$, $0 < m < 1$, then $\Omega(\eta) = sd(\eta)$, thus, we have

$$U_8 = \left( a_0 + a_2 \left( \frac{sd^2(\eta)}{f sd^2(\eta) + g} \right) \right) e^{i\theta(x,t)}, \tag{56}$$

where $f$ and $g$ are given as

$$f = \frac{h_4(2m^2 - h_2 - 1)}{(2m^2 - h_2 - 1)^2 - 2(2m^2 - 1)(2m^2 - h_2 - 1) - 3(1 - m^2)m^2},$$
$$g = \frac{3h_4}{(2m^2 - h_2 - 1)^2 - 2(2m^2 - 1)(2m^2 - h_2 - 1) - 3(1 - m^2)m^2}, \tag{57}$$

when $m \to 1$, $\Omega(\eta) = sd(\eta) = \sinh(\eta)$, we can acquire an SWS.

$$U_{8,1} = \left( a_0 + a_2 \left( \frac{\sinh^2(\eta)}{f \sinh^2(\eta) + g} \right) \right) e^{i\theta(x,t)}, \tag{58}$$

when $m \to 0$, $\Omega(\eta) = sd(\eta) = \sin(\eta)$, we obtain.

$$U_{8,2} = \left( a_0 + a_2 \left( \frac{\sin^2(\eta)}{f \sin^2(\eta) + g} \right) \right) e^{i\theta(x,t)}, \tag{59}$$

under constraint conditions defined as

$$\left( \frac{a_0 a_2 c}{2(a-4N)} \right)^2 \left( 2m^2 - \frac{P - a_0^2 c + ak^2 + w}{4(a-4N)} - 1 \right)$$
$$\times \left( \begin{array}{c} -\left( 2m^2 - \frac{P - a_0^2 c + ak^2 + w}{4(a-4N)} - 1 \right) \\ \left( 2(2m^2 - 1) + \frac{P - a_0^2 c + ak^2 + w}{4(a-4N)} \right) - 9(1 - m^2)m^2 \end{array} \right)$$
$$- \frac{3 a_2^2 c}{8(a-4N)} \left( -3(1 - m^2)m^2 - (2m^2 - 1)^2 + (\frac{P - a_0^2 c + ak^2 + w}{4(a-4N)})^2 \right)^2 = 0.$$

**Result 9**

If $l_0 = 1 - m^2$, $l_2 = 2 - m^2$, $l_4 = 1$, $0 < m < 1$, then $\Omega(\eta) = cs(\eta)$, thus, we have

$$U_9 = \left( a_0 + a_2 \left( \frac{cs^2(\eta)}{fcs^2(\eta) + g} \right) \right) e^{i\theta(x,t)}, \tag{60}$$

where $f$ and $g$ are given as

$$f = \frac{h_4(-m^2 - h_2 + 2)}{(-m^2 - h_2 + 2)^2 - 2(2 - m^2)(-m^2 - h_2 + 2) + 3(1 - m^2)},$$
$$g = \frac{3(1 - m^2)h_4}{(-m^2 - h_2 + 2)^2 - 2(2 - m^2)(-m^2 - h_2 + 2) + 3(1 - m^2)}, \tag{61}$$

when $m \to 1$, $\Omega(\eta) = cs(\eta) = \text{csch}(\eta)$, we can acquire an SWS.

$$U_{9,1} = \left( a_0 + a_2 \left( \frac{\text{csch}^2(\eta)}{f \text{csch}^2(\eta) + g} \right) \right) e^{i\theta(x,t)}, \tag{62}$$

when $m \to 0$, $\Omega(\eta) = sd(\eta) = \cot(\eta)$, we obtain.

$$U_{9,2} = \left( a_0 + a_2 \left( \frac{\cot^2(\eta)}{f \cot^2(\eta) + g} \right) \right) e^{i\theta(x,t)}, \tag{63}$$

under constraint conditions defined as

$$\left( \frac{a_0 a_2 c}{2(a - 4N)} \right)^2 \left( -m^2 - \frac{P - a_0^2 c + ak^2 + w}{4(a - 4N)} + 2 \right)$$
$$\times \left( \begin{array}{c} 9(1 - m^2) - \left( m^2 - \frac{P - a_0^2 c + ak^2 + w}{4(a - 4N)} + 2 \right) \\ \left( 2(2 - m^2) + \frac{P - a_0^2 c + ak^2 + w}{4(a - 4N)} \right) \end{array} \right)$$
$$- \frac{3a_2^2 c}{8(a - 4N)} \left( -(2 - m^2)^2 + 3(1 - m^2) + \left( \frac{P - a_0^2 c + ak^2 + w}{4(a - 4N)} \right)^2 \right)^2 = 0.$$

where $h_2$, $h_4$ and $h_6$ are given in Equation (23).

**Result 10**

If $l_0 = -m^2(1 - m^2)$, $l_2 = 2m^2 - 1$, $l_4 = 1$, $0 < m < 1$, then $\Omega(\eta) = ds(\eta)$, thus, we have

$$U_{10} = \left( a_0 + a_2 \left( \frac{ds^2(\eta)}{fds^2(\eta) + g} \right) \right) e^{i\theta(x,t)}, \tag{64}$$

where $f$ and $g$ are given as

$$f = \frac{h_4(2m^2 - h_2 - 1)}{(2m^2 - h_2 - 1)^2 - 2(2m^2 - 1)(2m^2 - h_2 - 1) + 3(1 - m^2)m^2},$$
$$g = \frac{3m^2(1 - m^2)h_4}{(2m^2 - h_2 - 1)^2 - 2(2m^2 - 1)(2m^2 - h_2 - 1) + 3(1 - m^2)m^2}, \tag{65}$$

when $m \to 1$, $\Omega(\eta) = ds(\eta) = \text{csch}(\eta)$, we can acquire an SWS.

$$U_{10,1} = \left( a_0 + a_2 \left( \frac{\text{csch}^2(\eta)}{f \text{csch}^2(\eta) + g} \right) \right) e^{i\theta(x,t)}, \tag{66}$$

when $m \to 0$, $\Omega(\eta) = ds(\eta) = \csc(\eta)$, we obtain.

$$U_{10,2} = \left( a_0 + a_2 \left( \frac{\csc^2(\eta)}{f \csc^2(\eta) + g} \right) \right) e^{i\theta(x,t)}, \tag{67}$$

under constraint conditions defined as

$$\left(\frac{a_0 a_2 c}{2(a-4N)}\right)^2 \left(2m^2 - \frac{P-a_0{}^2c+ak^2+w}{4(a-4N)} - 1\right)$$

$$\times \begin{pmatrix} -\left(2m^2 - \frac{P-a_0{}^2c+ak^2+w}{4(a-4N)} - 1\right) \\ \left(2(2-m^2) + \frac{P-a_0{}^2c+ak^2+w}{4(a-4N)}\right) - 9(1-m^2)m^2 \end{pmatrix}$$

$$-\frac{3a_2{}^2c}{8(a-4N)}\left(-3(1-m^2)m^2 - (2m^2-1) + \left(\frac{P-a_0{}^2c+ak^2+w}{4(a-4N)}\right)^2\right)^2 = 0.$$

**Result 11**

If $l_0 = \frac{1}{4}(1-m^2)$, $l_2 = \frac{1}{2}(m^2+1)$, $l_4 = \frac{1}{4}(1-m^2)$, $0 < m < 1$, then $\Omega(\eta) = nc(\eta) \pm sc(\eta)$ or $\frac{cn(\eta)}{1\pm sn(\eta)}$, thus, we have

$$U_{11} = \left(a_0 + a_2\left(\frac{(nc(\eta) \pm sc(\eta))^2}{f(nc(\eta) \pm sc(\eta))^2 + g}\right)\right)e^{i\theta(x,t)}, \tag{68}$$

where $f$ and $g$ are given as

$$f = \frac{h_4\left(\frac{1}{2}(m^2+1)-h_2\right)}{\left(\frac{1}{2}(m^2+1)-h_2\right)^2-(m^2+1)\left(\frac{1}{2}(m^2+1)-h_2\right)+\frac{3}{16}(1-m^2)^2},$$
$$g = \frac{3(1-m^2)h_4}{\left(\frac{1}{2}(m^2+1)-h_2\right)^2-(m^2+1)\left(\frac{1}{2}(m^2+1)-h_2\right)+\frac{3}{16}(1-m^2)^2}, \tag{69}$$

when $m \to 1$, $\Omega(\eta) = nc(\eta) \pm sc(\eta) = \cosh(\eta) \pm \sinh(\eta)$, we can acquire an SWS.

$$U_{11,1} = \left(a_0 + a_2\left(\frac{(\cosh(\eta) \pm \sinh(\eta))^2}{f(\cosh(\eta) \pm \sinh(\eta))^2 + g}\right)\right)e^{i\theta(x,t)}, \tag{70}$$

or $\Omega(\eta) = \frac{cn(\eta)}{1\pm sn(\eta)} = \frac{\text{sech}(\eta)}{1\pm\tanh(\eta)}$, we obtained

$$U_{11,2} = \left(a_0 + a_2\left(\frac{\text{sech}^2(\eta)}{f\,\text{sech}^2(\eta) + g(1 \pm \tanh(\eta))^2}\right)\right)e^{i\theta(x,t)}, \tag{71}$$

when $m \to 0$, $\Omega(\eta) = nc(\eta) \pm sc(\eta) = \sec(\eta) \pm \tan(\eta)$, we obtained

$$U_{11,3} = \left(a_0 + a_2\left(\frac{(\sec(\eta) \pm \tan(\eta))^2}{f(\sec(\eta) \pm \tan(\eta))^2 + g}\right)\right)e^{i\theta(x,t)}, \tag{72}$$

or $\Omega(\eta) = \frac{cn(\eta)}{1\pm sn(\eta)} = \frac{\cos(\eta)}{1\pm\sin(\eta)}$, we obtained.

$$U_{11,4} = \left(a_0 + a_2\left(\frac{\cos^2(\eta)}{f\cos^2(\eta) + g(1 \pm \sin(\eta))^2}\right)\right)e^{i\theta(x,t)}, \tag{73}$$

under constraint condition defined as

$$\left(\frac{a_0 a_2 c}{2(a-4N)}\right)^2 \left(\frac{1}{2}(m^2+1) - \frac{P-a_0{}^2c+ak^2+w}{4(a-4N)}\right)$$

$$\times \begin{pmatrix} \frac{9}{16}(1-m^2)^2 - \left(\frac{1}{2}(m^2+1) - \frac{P-a_0{}^2c+ak^2+w}{4(a-4N)}\right) \\ \left(m^2 + \frac{P-a_0{}^2c+ak^2+w}{4(a-4N)} + 1\right) \end{pmatrix}$$

$$-\frac{3a_2{}^2c}{8(a-4N)}\left(\frac{3}{16}(1-m^2)^2 - \frac{1}{4}(m^2+1)^2 + \left(\frac{P-a_0{}^2c+ak^2+w}{4(a-4N)}\right)^2\right)^2 = 0.$$

**Result 12**

If $l_0 = -\frac{1}{4}(1-m^2)^2$, $l_2 = \frac{1}{2}(m^2+1)$, $l_4 = -\frac{1}{4}$, $0 < m < 1$, then $\Omega(\eta) = ncn(\eta) \pm dn(\eta)$, thus, we have

$$U_{12} = \left( a_0 + a_2 \left( \frac{(ncn(\eta) \pm dn(\eta))^2}{f(ncn(\eta) \pm dn(\eta))^2 + g} \right) \right) e^{i\theta(x,t)}, \tag{74}$$

where $f$ and $g$ are given as

$$f = \frac{h_4\left(\frac{1}{2}(m^2+1)-h_2\right)}{\left(\frac{1}{2}(m^2+1)-h_2\right)^2 - (m^2+1)\left(\frac{1}{2}(m^2+1)-h_2\right)+\frac{3}{16}(1-m^2)^2},$$
$$g = \frac{3(1-m^2)h_4}{4\left(\left(\frac{1}{2}(m^2+1)-h_2\right)^2 - (m^2+1)\left(\frac{1}{2}(m^2+1)-h_2\right)+\frac{3}{16}(1-m^2)^2\right)}, \tag{75}$$

When $m \to 1$, $\Omega(\eta) = ncn(\eta) \pm dn(\eta) = n\text{sech}(\eta) \pm \text{sech}(\eta)$, we can acquire an SWS.

$$U_{12,1} = \left( a_0 + a_2 \left( \frac{(n\text{sech}(\eta) + \text{sech}(\eta))^2}{f(n\text{sech}(\eta) + \text{sech}(\eta))^2 + g} \right) \right) e^{i\theta(x,t)}, \tag{76}$$

when $m \to 0$, $\Omega(\eta) = ncn(\eta) \pm dn(\eta) = n\cos(\eta) \pm 1$, we obtained

$$U_{12,2} = \left( a_0 + a_2 \left( \frac{(n\cos(\eta) \pm 1)^2}{f(n\cos(\eta) \pm 1)^2 + g} \right) \right) e^{i\theta(x,t)}, \tag{77}$$

under constraint condition defined as

$$\left(\frac{a_0 a_2 c}{2(a-4N)}\right)^2 \left(\frac{1}{2}(m^2+1) - \frac{P-a_0^2c+ak^2+w}{4(a-4N)}\right)$$
$$\times \left( \begin{array}{c} \frac{9}{16}(1-m^2)^2 - \left(\frac{1}{2}(m^2+1) - \frac{P-a_0^2c+ak^2+w}{4(a-4N)}\right) \\ \left(m^2 + \frac{P-a_0^2c+ak^2+w}{4(a-4N)} + 1\right) \end{array} \right)$$
$$-\frac{3a_2^2c}{8(a-4N)}\left(\frac{3}{16}(1-m^2)^2 - \frac{1}{4}(m^2+1)^2 + \left(\frac{P-a_0^2c+ak^2+w}{4(a-4N)}\right)^2\right)^2 = 0.$$

**Result 13**

If $l_0 = \frac{1}{4}$, $l_2 = \frac{1}{2}(1-2m^2)$, $l_4 = \frac{1}{4}$, $0 < m < 1$, then $\Omega(\eta) = \frac{sn(\eta)}{1 \pm cn(\eta)}$, thus, we have

$$U_{13} = \left( a_0 + a_2 \left( \frac{sn^2(\eta)}{fsn^2(\eta) + g(1 \pm cn(\eta))^2} \right) \right) e^{i\theta(x,t)}, \tag{78}$$

where $f$ and $g$ are given as

$$f = \frac{h_4\left(\frac{1}{2}(1-2m^2)-h_2\right)}{\left(\frac{1}{2}(1-2m^2)-h_2\right)^2 - (1-2m^2)\left(\frac{1}{2}(1-2m^2)-h_2\right)+\frac{3}{16}},$$
$$g = \frac{3h_4}{4\left(\left(\frac{1}{2}(1-2m^2)-h_2\right)^2 - (1-2m^2)\left(\frac{1}{2}(1-2m^2)-h_2\right)+\frac{3}{16}\right)}, \tag{79}$$

when $m \to 1$, $\Omega(\eta) = \frac{sn(\eta)}{1 \pm cn(\eta)} = \frac{\tanh(\eta)}{1 \pm \text{sech}(\eta)}$, we can acquire an SWS.

$$U_{13,1} = \left( a_0 + a_2 \left( \frac{\tanh^2(\eta)}{f\tanh^2(\eta) + g(1 \pm \text{sech}(\eta))^2} \right) \right) e^{i\theta(x,t)}, \tag{80}$$

when $m \to 0$, $\Omega(\eta) = \frac{sn(\eta)}{1 \pm cn(\eta)} = \frac{\sin(\eta)}{1 \pm \cos(\eta)}$, we obtained

$$U_{13,2} = \left( a_0 + a_2 \left( \frac{\sin^2(\eta)}{f \sin^2(\eta) + g(1 \pm \cos(\eta))^2} \right) \right) e^{i\theta(x,t)}, \tag{81}$$

under constraint conditions defined as

$$\left( \frac{a_0 a_2 c}{2(a-4N)} \right)^2 \left( \frac{1}{2}(1 - 2m^2) - \frac{P - a_0^2 c + ak^2 + w}{4(a-4N)} \right)$$
$$\times \left( \frac{9}{16} - \left( \frac{1}{2}(1 - 2m^2) - \frac{P - a_0^2 c + ak^2 + w}{4(a-4N)} \right) \left( -2m^2 + \frac{P - a_0^2 c + ak^2 + w}{4(a-4N)} + 1 \right) \right)$$
$$- \frac{3a_2^2 c}{8(a-4N)} \left( -\frac{1}{4}(1 - 2m^2)^2 + \left( \frac{P - a_0^2 c + ak^2 + w}{4(a-4N)} \right)^2 + \frac{3}{16} \right)^2 = 0.$$

**Result 14**

If $l_0 = \frac{1}{4}$, $l_2 = \frac{1}{2}(m^2 + 1)$, $l_4 = \frac{1}{4}(1 - m^2)^2$, $0 < m < 1$, then $\Omega(\eta) = \frac{sn(\eta)}{cn(\eta) \pm dn(\eta)}$, thus, we have

$$U_{14} = \left( a_0 + a_2 \left( \frac{sn^2(\eta)}{f sn^2(\eta) + g(cn(\eta) \pm dn(\eta))^2} \right) \right) e^{i\theta(x,t)}, \tag{82}$$

where $f$ and $g$ are given as

$$f = \frac{h_4 \left( \frac{1}{2}(m^2+1) - h_2 \right)}{\left( \frac{1}{2}(m^2+1) - h_2 \right)^2 - (m^2+1) \left( \frac{1}{2}(m^2+1) - h_2 \right) + \frac{3}{16}(1-m^2)^2},$$
$$g = \frac{3h_4}{4 \left( \left( \frac{1}{2}(m^2+1) - h_2 \right)^2 - (m^2+1) \left( \frac{1}{2}(m^2+1) - h_2 \right) + \frac{3}{16}(1-m^2)^2 \right)}, \tag{83}$$

when $m \to 1$, $\Omega(\eta) = \frac{sn(\eta)}{cn(\eta) \pm dn(\eta)} = \frac{\tanh(\eta)}{\text{sech}(\eta) \pm \text{sech}(\eta)}$, we can acquire an SWS.

$$U_{14,1} = \left( a_0 + a_2 \left( \frac{\tanh^2(\eta)}{f \tanh^2(\eta) + g(\text{sech}(\eta) \pm \text{sech}(\eta))^2} \right) \right) e^{i\theta(x,t)}, \tag{84}$$

when $m \to 0$, $\Omega(\eta) = \frac{sn(\eta)}{cn(\eta) \pm dn(\eta)} = \frac{sin(\eta)}{\cos(\eta) \pm 1}$, we obtained

$$U_{14,2} = \left( a_0 + a_2 \left( \frac{\sin^2(\eta)}{f \sin^2(\eta) + g(\cos(\eta) \pm 1)^2} \right) \right) e^{i\theta(x,t)}, \tag{85}$$

under constraint conditions defined as

$$\left( \frac{a_0 a_2 c}{2(a-4N)} \right)^2 \left( \frac{1}{2}(m^2 + 1) - \frac{P - a_0^2 c + ak^2 + w}{4(a-4N)} \right)$$
$$\times \left( \begin{array}{c} \frac{9}{16}(1 - m^2)^2 - \left( \frac{1}{2}(m^2 + 1) - \frac{P - a_0^2 c + ak^2 + w}{4(a-4N)} \right) \\ \left( m^2 + \frac{P - a_0^2 c + ak^2 + w}{4(a-4N)} + 1 \right) \end{array} \right)$$
$$- \frac{3a_2^2 c}{8(a-4N)} \left( \frac{3}{16}(1 - m^2)^2 - \frac{1}{4}(m^2 + 1)^2 + \left( \frac{P - a_0^2 c + ak^2 + w}{4(a-4N)} \right)^2 \right)^2 = 0.$$

## 5. Graphical Demonstration and Explanation

This section displays the graphical presentation of the obtained solution and the influence of the fractional order parameter. Figures 1–3 demonstrate the 3D and contour graphs for different values of the fractional parameter $\alpha$ for the trigonometric function answers of Equation (26). Additionally, we explain the sensitive analysis of the complex Ginzburg–Landau equation in Kerr law media. One can notice in the plotted Figures 4–7 that the dynamical system is sensitive to initial conditions.

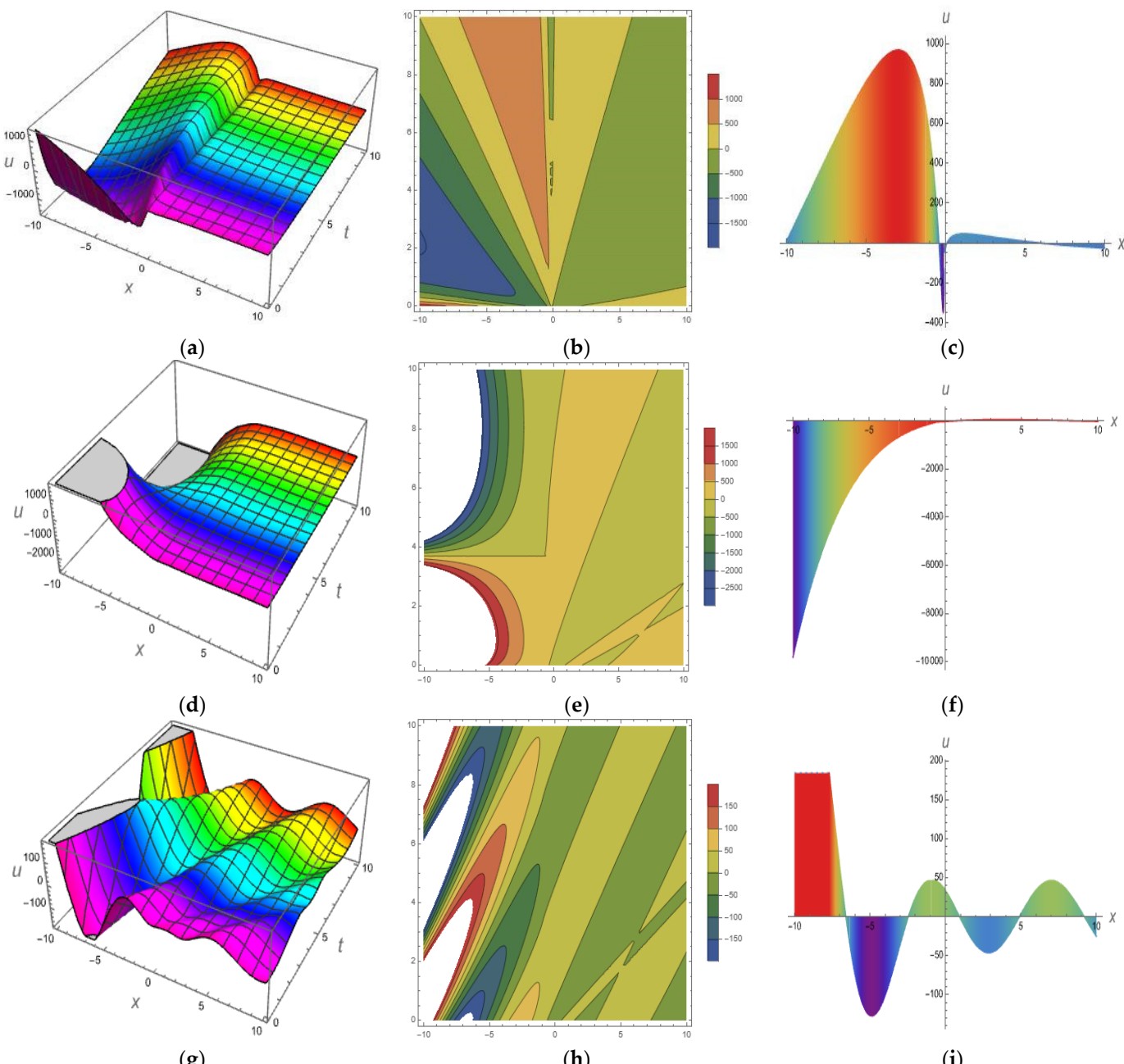

**Figure 1.** This figure presents the impact of fractional order on the solution $\mathrm{Re}(U_{1,1}(x,t))$ at $a = 1, w = 0.9, P = 0.9, k = 0.9, c = 1, N = 0.7, a_0 = 2, a_2 = 11.56449544I, v = 0.1$. (**a**) 3D visualization at. (**b**) Contour visualization at. (**c**) 2D visualization at $\alpha = 0.1$. (**d**) 3D visualization at $\alpha = 0.5$. (**e**) Contour visualization at $\alpha = 0.5$. (**f**) 2D visualization at $\alpha = 0.8$. (**g**) 3D visualization at $\alpha = 0.9$. (**h**) Contour visualization at $\alpha = 0.9$. (**i**) 2D Visualization at $\alpha = 0.9$.

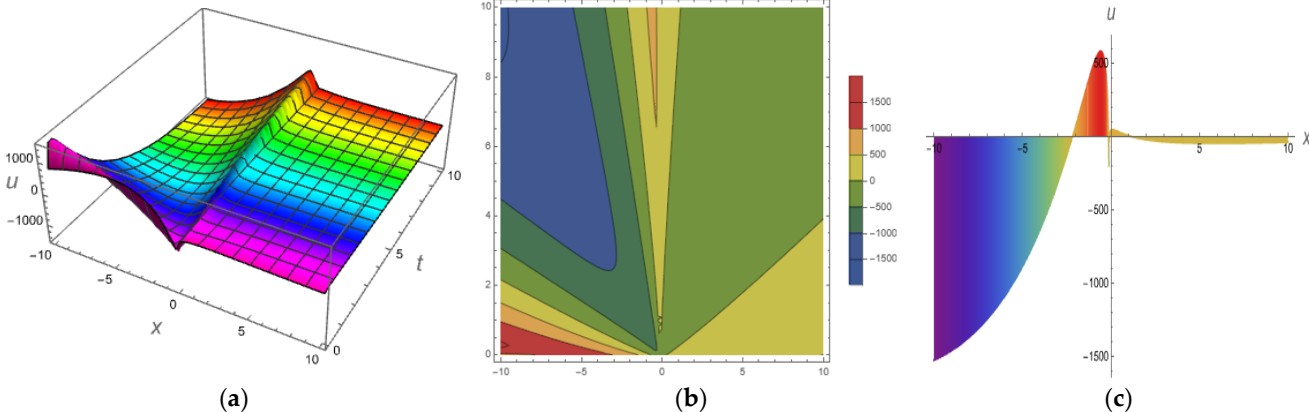

**Figure 2.** This figure presents the impact of fractional order on the solution $\text{Re}(U_{1,1}(x,t))$ at $a = 1, w = 0.9, P = 0.9, k = 0.9, c = 1, N = 0.7, a_0 = 2, a_2 = 11.56449544I, v = 0.1$. (**a**) 3D visualization at $\alpha = 0.95$. (**b**) Contour visualization at $\alpha = 0.95$. (**c**) 2D visualization at $\alpha = 0.95$. (**d**) 3D visualization at $\alpha = 0.99$. (**e**) Contour visualization at $\alpha = 0.99$. (**f**) 2D visualization at $\alpha = 0.99$.

**Figure 3.** *Cont.*

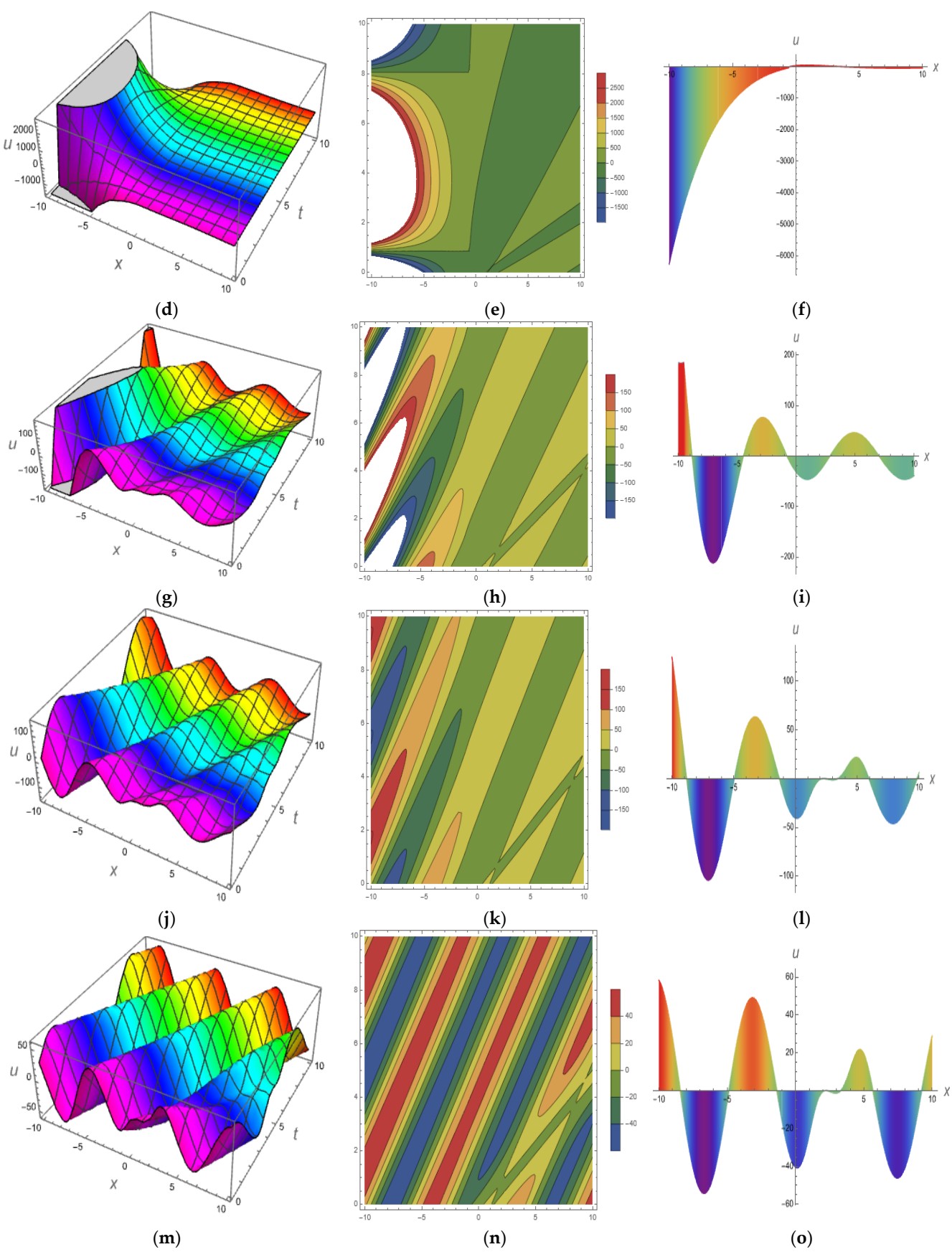

**Figure 3.** This figure presents the impact of fractional order on the solution Im $g(U_{1,1}(x, t))$ at $a = 1, w = 0.9, P = 0.9, k = 0.9, c = 1, N = 0.7, a_0 = 2, a_2 = 11.56449544I, v = 0.1$. (**a**) 3D Visualization

at $\alpha = 0.1$. (**b**) Contour visualization at $\alpha = 0.1$. (**c**) 2D visualization at $\alpha = 0.1$. (**d**) 3D visualization at $\alpha = 0.5$. (**e**) Contour visualization at $\alpha = 0.5$. (**f**) 2D visualization at $\alpha = 0.5$. (**g**) 3D visualization at $\alpha = 0.9$. (**h**) Contour visualization at $\alpha = 0.9$. (**i**) 2D visualization at $\alpha = 0.9$. (**j**) 3D visualization at $\alpha = 0.95$. (**k**) Contour visualization at $\alpha = 0.95$. (**l**) 2D visualization at $\alpha = 0.95$. (**m**) 3D visualization at $\alpha = 0.99$. (**n**) Contour visualization at $\alpha = 0.99$. (**o**) 2D visualization at $\alpha = 0.99$.

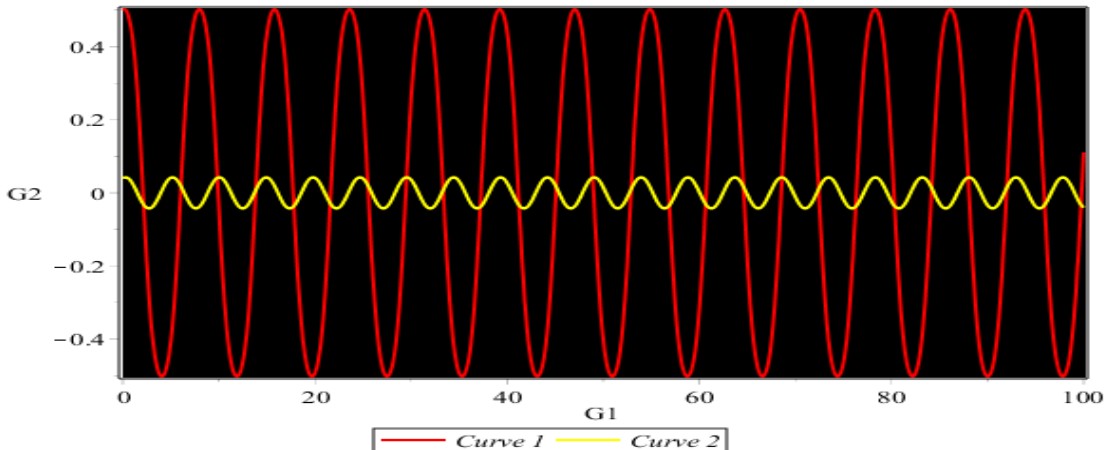

**Figure 4.** Sensitivity presentation for curve 1 at (0.5, 0.03) and curve 2 at (0.04, 0.02).

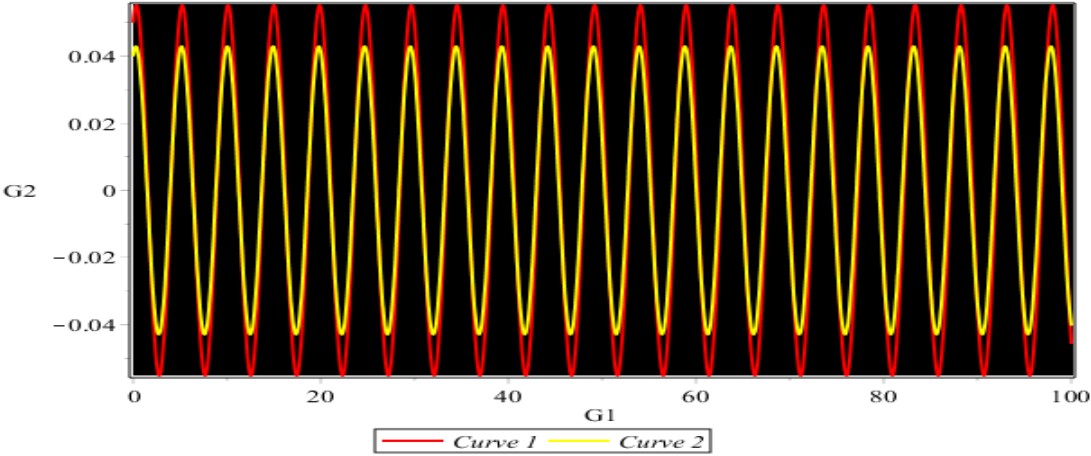

**Figure 5.** Sensitivity presentation for curve 1 at (0.05, 0.03) and curve 2 at (0.04, 0.02).

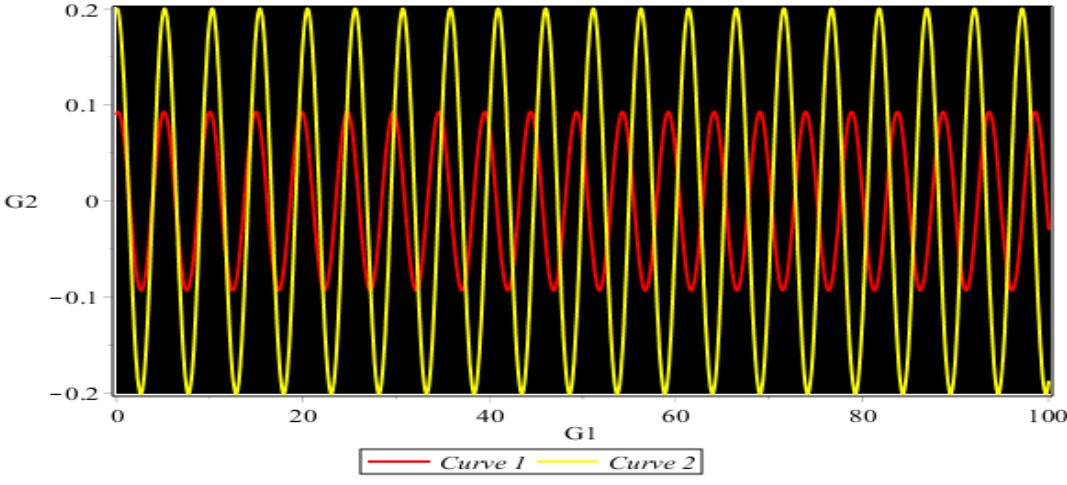

**Figure 6.** Sensitivity presentation for curve 1 at (0.09, 0.03) and curve 2 at (0.2, 0.02).

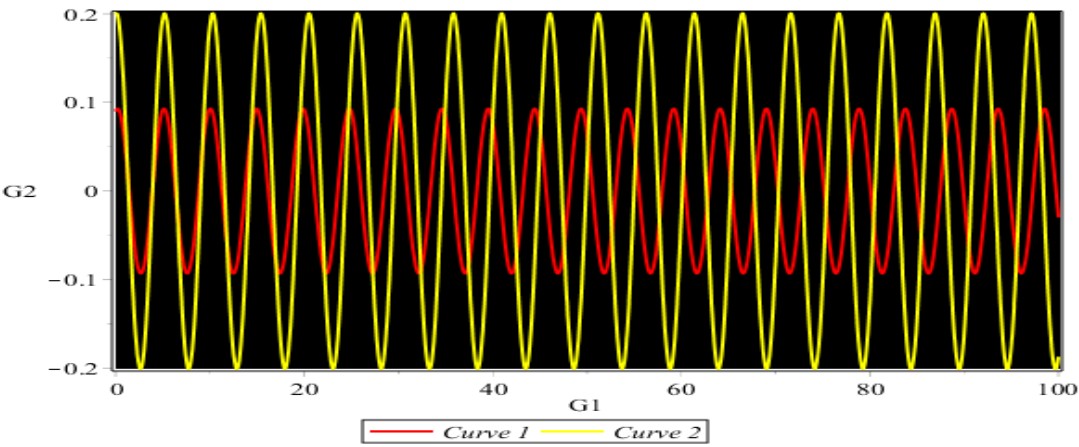

**Figure 7.** Sensitivity presentation for curve 1 at (0.04, 0.03) and curve 2 at (0.4, 0.02).

## 6. Sensitive Visualization

In this section, the sensitivity analysis is performed to investigate the sensitivity of the complex Ginzburg-Landau equation in Kerr law media using the computational software MAPLE. The Galilean transformation is applied to the ODE and obtains a dynamical system as given below.

Let $u(\eta) = G_1(\eta), \ u\prime(\eta) = G_2(\eta),$

$$\frac{dG_1}{d\eta} = G_2,$$
$$\frac{dG_2}{d\eta} = -\frac{w + ak^2 + P}{a - 4N}G_1{}^3 + \frac{c}{a - 4N}G_1.$$

The figures mentioned above are plotted to visualize the sensitive behavior of the dynamic system at $a = 0.2, \ w = 0, \ P = 1, \ k = 0.5, \ c = 1, \ N = 0.2.$

We must perform a sensitivity analysis to determine how sensitive our system is. If only a slight modification is made to the initial conditions, the system's sensitivity will be inferior. The system will be pretty sensitive if small changes in the initial conditions cause a significant shift. Many graphs are constructed for various initial condition values to demonstrate the system's sensitivity. One can notice from the above-plotted figures that; the dynamical system is subtle concerning initial circumstances.

## 7. Conclusions

In this research, we have used the beta-derivative to find the exact solutions of the fractional CGLE. We conceded this objective by assuming a particular wave transformation to adjust the fractional CGLE to a nonlinear ODE of second order such that the resultant ODE could be resolved by engaging the $\phi^6$-model expansion method. This method restored the periodic, dark, bright, dark-bright, exponential, trigonometric, and rational solitons for Kerr law non-linearity. To designate the physical phenomena of the space-time fractional CGLE, some solutions are produced in shape by allocating values to parameters in 3D under some particular constraints. Comparing other work [38,42–49], our solution was not described in prior works. Additionally, these systems are very operative and potent in finding soliton solutions of nonlinear fractional differential equations, and the solutions gained can support us in designating the nonlinear dynamics of optical soliton propagations in more penetration.

The findings are listed below:

- There are 28 analytical solutions discovered with fourteen distinct families.
- The acquired wave patterns are based on Jacobi elliptic functions, with hyperbolic solutions obtained for limiting case $m \to 1,$ and trigonometric solutions developed for limiting case $m \to 0.$

- Every obtained traveling wave solution has a related condition constructed to guarantee the existence of the solution.
- On suitable values of the involved parameters, which satisfy the specified constraints, 3D and contour real and imaginary profiles of the solutions are shown.
- The fractional order parameter is responsible for controlling the singularity of the soliton solution.
- The sensitivity analysis ensures that the model is sensitive to initial conditions.

This method can be applied to many NLPDEs in mathematical physics. Finally, our solutions have been checked using MATHEMATICA by putting them back into the original equation.

**Author Contributions:** Conceptualization, R.M.Z. and W.-X.M.; methodology, W.-X.M.; software, W.A.F.; validation, S.M.E. and W.A.F.; formal analysis, S.M.E. and K.B.M.; investigation, K.B.M.; resources, W.A.F.; data curation, W.A.F.; writing—original draft preparation, R.M.Z. and K.B.M.; writing—review and editing, R.M.Z. and W.-X.M.; visualization, S.M.E.; supervision, W.-X.M.; project administration, W.-X.M.; funding acquisition, W.-X.M. and S.M.E. All authors have read and agreed to the published version of the manuscript.

**Funding:** The work was supported in part by NSFC under the grants 12271488, 11975145 and 11972291, the Ministry of Science and Technology of China (G2021016032L), and the Natural Science Foundation for Colleges and Universities in Jiangsu Province (17 KJB 110020).

**Data Availability Statement:** Not applicable.

**Acknowledgments:** This work was partially funded by the research center of the Future University in Egypt, 2022.

**Conflicts of Interest:** The authors declare that they have no competing interests.

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
