# Peer review of "New Explicit Propagating Solitary Waves Formation and Sensitive Visualization of the Dynamical System"

_fractalfract, doi:10.3390/fractalfract7010071_

Round 1

Reviewer 1 Report

It is interesting that the paper extends the \phi-6 model expansion method the \bate-time frational CGL equation and obtains many exact solutions. The paper is good but should be improved accordingly the comments:

1) In fact, the \phi-6 model expansion method is special cases of the generalized F-expansion method and the generalized auxiliary equation method proposed and applied in the following papers: Sheng Zhang and Tiecheng Xia, A generalized auxiliary equation method and its application to (2+1)-dimensional asymmetric Nizhnik-Novikov-Vesselov equationsï¼›J. Phys. A: Math. Theor. 40 (2) (2007) 227-248, this should be introduced and cited. More importantly, the 14 special solutions to the \phi-6 model (9) should be cited in Zhang et als papers, for example, Sheng Zhang and Tiecheng Xia, A generalized F-expansion method and new exact solutions of Konopelchenko-Dubrovsky equations, Appl. Math. Comput. 183(2) (2006) 1190-1200, which were published before [55-59] as cited by authors. There [59] is missing.

2) Several typos should be corrected.

Author Response

First of all, we would like to express our sincere gratitude to the reviewers for their helpful comments and suggestions in improving the quality of this paper. We have revised the manuscript according to the reviewers’ comments. In the following, we present replies to reviewers’ suggestions one by one (our replies are marked in blue).

To Reviewer #1:

It is interesting that the paper extends the \phi-6 model expansion method the \bate-time frational CGL equation and obtains many exact solutions. The paper is good but should be improved accordingly the comments:

  • In fact, the \phi-6 model expansion method is special cases of the generalized F-expansion method and the generalized auxiliary equation method proposed and applied in the following papers: Sheng Zhang and Tiecheng Xia, A generalized auxiliary equation method and its application to (2+1)-dimensional asymmetric Nizhnik-Novikov-Vesselov equationsï¼› Phys. A: Math. Theor. 40 (2) (2007) 227-248, this should be introduced and cited. More importantly, the 14 special solutions to the \phi-6 model (9) should be cited in Zhang et al’s papers, for example, Sheng Zhang and Tiecheng Xia, A generalized F-expansion method and new exact solutions of Konopelchenko-Dubrovsky equations, Appl. Math. Comput. 183(2) (2006) 1190-1200, which were published before [55-59] as cited by authors. There [59] is missing.

Response: Introduction has been improved by adding the said recent papers in references. Please see the edit marked in yellow on Page 26, references [23, 24].

2) Several typos should be corrected.

Response: Authors have properly proofread the paper, and typos errors have been removed. Please see the revised manuscript. 

Author Response

First of all, we would like to express our sincere gratitude to the reviewers for their helpful comments and suggestions in improving the quality of this paper. We have revised the manuscript according to the reviewers’ comments. In the following, we present replies to reviewers’ suggestions one by one (our replies are marked in blue).

  1. The original contributions need to be much better presented in the last paragraph of section \INTRODUCTION. All improvements, if they are, and new results must be described in this paragraph. The advantages of the work are not discussed in the text.

Response: The original contributions, improvements, and advantages of the work have been added in the second last paragraph of the Introduction. Please see the edit marked in yellow on Page 3.

  1. A brief description of the structure (layout) of the paper may be added to the end of the Introduction.

Response: A brief description of the structure (layout) of the paper has been added at the end of the Introduction. Please see the edit marked in yellow on Page 3.

  1. The Abstract should be modified by adding advantages of the proposed method.

Response: Abstract has been modified by adding the advantages of the proposed method. Please see the edit marked in yellow on Page 1.

  1. Check the manuscript carefully for typos and grammatical errors.

Response: Authors have properly proofread the paper, and typos and grammatical errors have been removed.

  1. Please clarify the novelty of this paper with respect to the published papers.

Response: The novelty of this paper with respect to the published papers has been added in the second last paragraph of the Introduction. Please see the edit marked in yellow on Page 3.

  1. The references list is not at all updated with latest developments and publications. I suggest the authors to keep up to date with the relevant literature such as

https://doi.org/10.1016/j.apnum.2022.07.018

https://doi.org/10.22436/jmcs.030.01.07

http://dx.doi.org/10.22436/jmcs.024.02.07

http://dx.doi.org/10.22436/jmcs.027.01.03                        

Response: The said relevant references had been added to the literature. Please see the edit marked in yellow on Page 26; please see the references [25, 26, 27] in the revised manuscript.

  1. In general, the typeset equations should be regarded as parts of a sentence and treated accordingly with the appropriate grammatical convention and punctuation. More editing for writing is needed. At the end of all equations must be put \COMMA" or \POINT" according to the typing rules.

Response: All the commas and full stops have been checked.

  1. All acronyms should be defined before.

Response: All acronyms have been defined before.

  1. Please rewrite the paper in the template of the journal. The references of this paper should be rewritten accordingly to style of the journal.

Response: Paper has been written in the template of the journal. Also, references have been written according to journal guidelines.

  1. Section Conclusion should be elaborated more in detail.

Response: The conclusion section has been elaborated in more detail. Please see the edit marked in yellow on Page 29.

  1. The mathematical results and the equations should be double-checked.

Response: The mathematical results and the equations have been double-checked.

  1. What is the advantage the local fractional derivative with respect to other definitions of fractional derivative?

Response: The local fractional derivative shows damping behavior and is reliable in finding the solutions of fractional DEs.

  1. Please interpret the obtained numerical results in different tables. It is important what you conclude to them.

Response: Numerical results have been concluded in sections 6 and 7 in the Introduction.

  1. At the beginning of the numerical results section, authors should present the configuration of the personal computer used to perform the simulation results.

Response: Done. Please see the edit marked in yellow on Pages 27 and 30.

  1. It seems that your paper has a very weak relationship with the field of Fractal Fract. Are you able to demonstrate the novelty of your work in comparison with some specific research findings already published in the Journal of Fractal Fract?

Response: Respected Sir, the following manuscript related to our work has been published in the journal of Fractal Fractional, so we claim that our work is within the scope of the Fractal Fractional journal.

  1. Exact Fractional Solution by Nucci’s Reduction Approach and New Analytical Propagating Optical Soliton Structures in Fiber-Optics, Fractal Fract. 2022, 6, 654. https://doi.org/10.3390/fractalfract6110654
  2. Exact Solutions of the Nonlinear Modified Benjamin-Bona-Mahony Equation by an Analytical Method. Fractal Fract. 2022, 6, 399. https://doi.org/10.3390/fractalfract6070399
  • Exact Traveling Wave Solutions of the Local Fractional Bidirectional Propagation System Equations. Fractal Fract. 2022, 6, 653. https://doi.org/10.3390/fractalfract6110653

  1. Please mention the digits of the Program used in numerical examples.

Response: Done. Please see the sections 6 and 7.

  1. Regarding the cases studied in the paper, what is the physics behind these choices.

Response: Physics behind the choices has been explained in the introduction section and in the sec. 7.

  1. Physical interpretation is missed.

Response: Physical interpretation has been explained in the Secs. 1
